# Comparing Depression Prevalence and Associated Symptoms with Intolerance of Uncertainty among Chinese Urban and Rural Adolescents: A Network Analysis

**DOI:** 10.3390/bs13080662

**Published:** 2023-08-08

**Authors:** Zhi Jing, Fengqin Ding, Yishu Sun, Sensen Zhang, Ning Li

**Affiliations:** 1Department of Psychology, College of Teacher Education, Ningxia University, Yinchuan 750021, China; nickolastin@oit.edu.cn (Z.J.);; 2Department of Mental Health Education, Office of Student Affairs, Ordos Institute of Technology, Ordos 017000, China; 3Psychotherapy Department, Ningxia Mental Health Center, Yinchuan 750021, China

**Keywords:** intolerance of uncertainty, depression, rural adolescents, network analysis

## Abstract

The prevalence of depression among adolescents is increasing, which can hinder their healthy development and is intricately linked to the intolerance of uncertainty (IU). IU involves both prospective anxiety and inhibitory anxiety. However, the precise relationship between depressive symptoms and these two components of IU remains unclear, particularly when considering the specific context of rural adolescents in China. A total of 1488 adolescents (male, 848; *Mean*_age_ = 20, *SD*_age_ = 1.51, age range from 16 to 24) in China were recruited and divided into urban adolescents (N = 439) and rural adolescents (N = 1049) groups. The Patient Health Questionnaire-9 and Intolerance of Uncertainty Scale-12 were utilized to measure depression and IU. The symptom network approach and the flow network approach were employed. The prevalence of depression was significantly higher (*χ*^2^ = 4.09, *p* = 0.04) among rural adolescents (N = 419, 40.1%) than urban adolescents (N = 152, 34.8%). The node strength of “motor” demonstrated some discrepancy between rural and urban adolescents, while there was no notable disparity in the global strength and structure of the network between the two groups. However, rural adolescents exhibited a significantly higher global strength in the flow network (including depression and IU) than their urban counterparts. In the flow networks of rural adolescents, “guilt” was directly associated with prospective and inhibitory anxiety. These findings highlight the urgent need for interventions that enhance the ability of rural adolescents to cope with uncertainty and prevent their depressive symptoms more effectively.

## 1. Introduction

Depression is a common mental disorder that negatively influences psychosocial function and quality of life, and was ranked as the third cause of disease burden worldwide by the WHO [1]. Noticeably, depression is prevalent among adolescents, and a meta-analysis found that the prevalence rate reached 29% (95% *CI*: 17%, 40%) [2]. Moreover, depression can hamper the healthy development of adolescents. According to a previous study, depression may lead to the impairment of the central executive process, leading to poorer academic performance [3]. Meanwhile, adolescents suffering from depression tend to show more problem behaviors and more social problems [4], and more alarmingly, depression was found to serve as an important risk factor for suicide attempts and suicidal ideas among adolescents [5]. Indeed, to provide evidence for designing effective interventions, it is crucial to further explore the underlying mechanism of depression among adolescents.

Notably, educational resources, community structure, and economic development may show great diversity between rural and urban areas in China [6,7]. Compared to the residents in urban areas, people living in rural areas have poorer access to professional and effective mental support resources [8]. As the ecological system theory states, individual development depends highly on social surroundings and interactions with them [9]. Considering the divergent social surroundings in rural and urban areas and the fact that adolescents are sensitive to the influence of social surroundings [10], the prevalence rate and symptomatic character of depression may vary across rural and urban areas. A study has identified some distinctions in depression between adolescents in rural areas and their peers in urban areas. It found that adolescents with poor academic achievements or females were at a higher risk of suffering from depression, specifically in rural areas [11]. Despite the strong association between lower socio-economic status and depression [12], previous studies have predominantly treated adolescents as a homogeneous group, neglecting to explore potential differences among subgroups based on varying socio-economic statuses. This oversight may impede our understanding and hinder the development of effective treatment strategies for depression. Hence, it is urgently warranted to explore the character of depression between rural and urban adolescents and provide more efficient mental support services in rural areas.

Several risk factors related to depression have been identified, including low family income, negative school experience [13], peer victimization [14], and intolerance of uncertainty [15]. Intolerance of uncertainty (IU), one of the risk factors for depression [16], is characterized as “the tendency to react negatively on an emotional, cognitive, and behavior level to uncertainty situations and events” [17], which consists of two components, namely prospective anxiety and inhibitory anxiety [18]. Prospective anxiety refers to fear and anxiety in the prospect of uncertainty, while inhibitory anxiety relates to avoidance or inaction when confronting uncertainty [16]. According to the stress theory proposed by Lazarus [19], stress is defined as the interaction between individuals and their environment, and stress may appear when a person perceives the environment as harmful. This suggests that the experience of stress is heavily influenced by how people interpret the situations they encounter. Therefore, since people with higher levels of IU may tend to overestimate the likelihood of negative events when confronting uncertainty, they are more likely to become more stressed when confronting unpredictable situations [15,20]. Similarly, the IU model proposed by Dugas et al. [21] suggested that people with high IU tended to be stressed by ambiguous cues and might have difficulty coping with negative events.

Indeed, IU may serve as a transdiagnostic risk and maintaining factor for depression [15,16]. In a comprehensive study examining the impact of multiple factors, such as IU, meaning in life, and chronic illness, on depression and anxiety in the general population, it was revealed that IU emerged as a significant predictor of both depression and anxiety, surpassing other factors in importance [22]. Likewise, another study indicated that mitigating IU during the COVID-19 pandemic could alleviate depression, stress, and anxiety while promoting a more positive outlook, including increased optimism and positive emotional states [20]. As mentioned before, IU comprises two components, namely prospective anxiety and inhibitory anxiety [18], and limited studies have looked into the diverse relationship between the two components of IU and depression. A study suggested that, compared to prospective anxiety, inhibitory anxiety is more closely connected with depression [16], which is consistent with the finding that people suffering from depression tend to show more avoidant behaviors [23]. Furthermore, another study also indicated that inhibitory anxiety predicted changes in anxiety and depression symptoms, while the reduction of prospective anxiety predicted the improvement of the overall mental health state [24]. This finding provides further evidence for the potential hypothesis that the two components of IU may influence depression differently. Despite all the studies, the underlying mechanism between IU and depression symptoms remains unclear, which warrants further study.

The traditional diagnostic perspective tended to treat mental disorders as existing entities may ignore the fact that mental disorders are comprised of various symptoms and may interact with each other [25]. Although previous studies have proved that IU is closely related to depression [15,16,20,22], they failed to identify how two components of IU interact with different depression symptoms specifically. This deficit may hinder the process of understanding and designing suitable treatments for people with higher levels of IU and depression. Compared to the traditional perspective, network analysis assumes that mental disorders can be defined as the combination of various symptoms and that symptoms may be triggered by other symptoms [26]. Therefore, by analyzing the correlation between different symptoms, the network analysis could identify the most influential symptom, which offers evidence for designing more effective interventions [26]. Hence, network analysis focused on the interaction between IU and depression symptoms is of great necessity.

To date, very limited studies use network analysis to identify the relationship between depression and IU. A study explored the interaction among depression, anxiety, and IU during COVID-19. In this study, depression and anxiety symptoms were more strongly correlated among people with high IU [27]. Consistent with the previous research, IU may play an important role in maintaining depression and anxiety. Another network study identified that “fatigue” and “frustration when facing uncertainty” were the two most central symptoms, and “frustration when facing uncertainty” was a bridge symptom connecting depression and IU [28]. The results showed that the symptom of IU, “frustration when facing uncertainty”, deserved more attention when designing interventions aimed at fostering the ability to cope with uncertainty. However, since there are two dimensions in the Intolerance of Uncertainty-12 (IU-12), named prospective anxiety and inhibitory anxiety [18], and each dimension may influence depression differently [16], whether it is reasonable to view each item in IU-12 as a separate symptom in the network analysis is debatable. Moreover, considering this study did not analyze the rural and urban participants separately, further studies looking into the possibly distinctive influence of two components in IU in rural and urban adolescents are warranted.

Hence, our study has three specific aims. First, we aim to compare the prevalence rate of depression among adolescents in rural and urban areas. Second, we want to identify the various possible network structures of depression in urban and rural adolescents through symptom networks. Finally, we aim to explore the interaction between the two components of IU (Intolerance of Uncertainty) and depression among adolescents from urban and rural areas using the flow network approach.

According to the aim of our study, we propose the following hypothesis.

**Hypothesis 1:** *The prevalence rate of depression in rural adolescents may be higher than in urban adolescents*.

**Hypothesis 2:** 
*Based on that, diverse sociodemographic factors influence rural areas compared to their urban counterparts. Therefore, we propose the symptom network of depression could differ between rural and urban adolescents.*


**Hypothesis 3a:** 
*In the flow network, we hypothesize that the symptom network of depression and IU could differ between rural and urban adolescents.*


**Hypothesis 3b:** 
*In the flow network, we hypothesize that inhibitory anxiety is directly linked to a greater number of depression symptoms in both rural and urban adolescents than prospective anxiety.*


## 2. Methods

### 2.1. Participants

In the present study, a written questionnaire was administered to undergraduate students enrolled in two applied universities in the western region of China in April 2023. The sample comprised students across all four years of study. A total of 1520 questionnaires were gathered. After considering that certain participants’ data on age did not align with the typical youth age range (i.e., 10–24 years old, [29]), a total of 32 data points were eliminated (e.g., there are 15 participants without age information, 15 participants over 24 years old, and two participants whose age is given as 2 years old.). The final data analysis comprised a total of 1488 questionnaires (female = 640; *Mean*_age_ = 20, *SD*_age_ = 1.51, range_age_ 16–24). According to the students’ household registration, the students were divided into urban (N = 439, female, 198; *Mean*_age_ = 20, *SD*_age_ = 1.54, range_age_ 17–24) and rural (N = 1049, female, 442; *Mean*_age_ = 20, *SD*_age_ = 1.50, range_age_ 16–24) groups. Considering the difference in population size between urban and rural areas, *t*-tests and Chi-square tests were conducted to examine the differences in age and gender between participants in the two groups. The results indicated that there were no significant differences in age (*t* = 0.49, *p* = 0.623, *Cohen’s d* = 0.03) and gender (*χ*^2^ = 1.11, *p* = 0.29) between the two groups.

The present investigation obtained ethics approval from the Science and Technology experimental ethics committee of Ningxia University (Number: NXU-23-051), and written informed consent was obtained from all participants before being involved.

### 2.2. Measures

#### 2.2.1. Patient Health Questionnaire-9 (PHQ-9)

The PHQ-9 is composed of nine items, as developed by Kroenke et al. [30]. The assessment of depressive symptoms involves grading each item on a scale ranging from 0 (indicating the absence of symptoms) to 3 (indicating the presence of symptoms nearly every day). The symptoms evaluated include anhedonia, sad mood, sleep, energy, appetite, guilt, concentration, motor function, and suicide ideation, all of which were assessed for the previous two weeks. Elevated scores indicate higher degrees of severity in depression. Previous studies have demonstrated the PHQ-9 was valid [31]. In the present study, PHQ-9 was found to be highly reliable (*α* = 0.94).

#### 2.2.2. Intolerance of Uncertainty Scale-12 (IUS-12)

IUS is a 27-item questionnaire designed to assess general reactions to uncertainty [32]. A shorter version of the IUS, known as the IUS-12, was used in this study and is highly correlated with the original 27-item version [33]. The IUS-12 utilizes a 5-point Likert scale, ranging from 1 (not at all characteristic of me) to 5 (entirely characteristic of me), and has been shown to have two factors, prospective anxiety and inhibitory anxiety. The Chinese version of the IUS-12 has demonstrated good reliability and validity [34], with a Cronbach’s *α* coefficient of 0.89 in the present study.

### 2.3. Data Analysis

All analyses were performed in R software (version 4.3.0, The R Foundation for Statistical Computing, Vienna, Austria). *Mean*, standard deviation (*SD*), skewness, and kurtosis of all PHQ-9 item scores and two dimensions of IUS-12 were inspected.

#### 2.3.1. Network Estimation

To estimate the network, Chen and Chen [35] and Epskamp et al. [36] proposed an extended Bayesian information criterion (EBIC) graphical least absolute shrinkage and selection operator (LASSO) model. The network consisted of nodes that represented individual symptoms, and the edges indicated the correlations between symptom pairs. Modifications were made to the correlation matrix to enhance clarity and simplicity, resulting in a more concise and interpretable representation of the network. Red edges represented negative correlations, while blue edges depicted positive correlations. The network estimation and visualization were carried out using the R packages *qgraph* 1.6.9 and *bootnet* 1.4.3, developed by Epskamp et al. [37] and Epskamp et al. [36], respectively. The *flow* graphical function was also employed to examine the symptoms directly related to suicidal ideation. This function constructed a vertical network by placing suicidal ideation on the left and displaying both direct and indirect symptoms in the network.

To assess the importance of nodes in our network, we utilized the centrality plot function from the *qgraph* package [37] to estimate three widely used node centrality indices: betweenness, closeness, and strength. According to the method Opsahl et al. [38] described, these indices assist in determining the importance of the network’s nodes. However, it is worth noting that traditional centrality indices such as strength may not provide accurate predictions of a node’s influence when the network contains both positive and negative edges [39]. To address this limitation, we employed ExpectedInfluence (EI) proposed by Robinaugh et al. [39]. By considering both positive and negative edges, the EI allowed us to evaluate the significance of each symptom more comprehensively. Furthermore, we assessed the predictability of the network using the *R*^2^ metric. This estimation was performed using the *mgm* package version 1.2–12 [40]. The *R*^2^ metric provided insights into the extent to which the network could explain the observed data, allowing us to evaluate its overall predictive performance.

#### 2.3.2. Network Stability and Accuracy

We employed bootstrapped confidence intervals (95% *CI*s) to assess the accuracy of the edges within the network. According to Epskamp et al. [36], narrower *CI*s indicate higher accuracy in the estimated network structure. We utilized the case-dropping bootstrap procedure to evaluate the stability of centrality indices, particularly strength, and EI, which generated the correlation stability coefficient (*CS-C*) [41]. The *CS-C* represents the maximum proportion of cases that can be dropped from the sample while still maintaining a correlation coefficient of at least 0.7 with the original sample’s centrality indices, with 95% probability. Generally, a *CS-C* value of 0.25 or higher is considered acceptable, while a value of 0.5 or higher is preferred. We also conducted bootstrap tests (1000 iterations) using 95% *CI*s to analyze differences in edge weights and centrality indices. If the *CI*s did not include zero, it indicated a statistically significant difference between two edges or two nodes.

All of these analyses were performed using the *bootnet* package (version 1.4.3) in R, developed by [36].

#### 2.3.3. Network Comparison

We utilized the *NetworkComparisonTest* package (version 2.2.1) in R, developed by Van Borkulo et al. [42], to perform the network comparison test (NCT). This test allowed us to assess the variation in edge stability (distributions of edge weights) and overall strength (sum of all edge weights) between the urban and rural group networks.

## 3. Results

### 3.1. Prevalence of Depression and Descriptive Statistics

According to previous studies [43], the optimal cut-off for identifying depression using the PHQ-9 questionnaire is a score of 8 or higher. Using this criterion, in the overall sample of adolescents, 571 participants met the criteria for depression (38.5%). Among them, the prevalence of depression was significantly higher (*χ*^2^ = 4.09, *p* = 0.04) in rural adolescents (N = 419, 40.1%) compared to urban adolescents (N = 152, 34.8%).

The abbreviations of the depressive symptoms, two dimensions of IU, and the *mean*, standard deviation (*SD*), skewness, and kurtosis of depressive symptoms measured by the PHQ-9 are presented in Table 1.

### 3.2. Symptom Network and Flow Network Structures

The depression symptom networks are shown in Figure 1. Appendix A and Part A of Figure 1 present the edge weights among all adolescents. The top three strongest edges are Anhedonia—Sad Mood (PHQ1–PHQ2), Anhedonia—Energy (PHQ1–PHQ4), and Motor—Suicide (PHQ8–PHQ9). The edge of Anhedonia and Sad Mood (PHQ1—PHQ2) demonstrates the strongest association, followed by Anhedonia–Energy (PHQ1–PHQ4) and Sleep—Energy (PHQ3–PHQ4) among urban adolescents, is shown in Appendix A and Part B of Figure 1. In contrast to urban adolescents, the top three strongest edges among rural adolescents are Anhedonia–Sad Mood (PHQ1–PHQ2), Anhedonia—Energy (PHQ1–PHQ4), and Motor—Suicide (PHQ8–PHQ9), which aligns with the findings of the entire adolescent sample, and is shown in Appendix A and Part C of Figure 1.

In Part D of Figure 1, “energy” and “motor” exhibited both the highest node strength in the depressive symptom network across all adolescents, urban adolescents, and rural adolescents. Despite this, after standardizing the scores, it was still observed that the magnitude of “motor” symptoms in urban areas was higher than that in rural areas, while the magnitude of “motor” symptoms in rural areas was higher than that in urban areas.

For the *flow* network structure, there were no indirect associations between depressive symptoms and prospective or inhibitory anxiety among all adolescents (See Parts A and B of Figure 2). Specifically, we observed that Guilt (PHQ6) was the most prominent symptom in the depression network, directly associated with both prospective and inhibitory anxiety. However, among urban adolescents, Appetite (PHQ5) indirectly correlated with prospective and inhibitory anxiety, and Concentration (PHQ7) and Anhedonia (PHQ1) emerged as the strongest symptoms in the depression network associated with prospective anxiety and inhibitory anxiety, respectively (See Part C of Figure 2). At the same time, Anhedonia (PHQ1) was the strongest symptom associated with inhibitory anxiety (See Part D of Figure 2). Among rural adolescents, no symptoms were indirectly associated with prospective anxiety, with Guilt (PHQ6) being the strongest associated symptom (See Part E of Figure 2). In terms of inhibitory anxiety, Anhedonia (PHQ1) had an indirect association, and Guilt (PHQ6) was the strongest symptom directly linked to it (See Part F of Figure 2).

### 3.3. Network Stability and Accuracy

In Figure 3, the case-dropping bootstrap procedure indicates that *CS-C* was 0.75, 0.52, and 0.75 among all, urban, and rural adolescents, respectively. The bootstrapped 95% confidence intervals (*CI*s) were narrow, indicating reliable edges (Appendix A). Furthermore, the nonparametric bootstrap procedure results demonstrated that most node strength and edge weights comparisons were statistically significant (Appendix A).

### 3.4. Network Comparison between Urban and Rural Adolescents

The Network Comparison Test found that no significant global network strength was observed between urban and rural adolescents (4.08 vs. 4.28, with a non-significant global strength difference (*S* = 0.20, *p* = 0.24), see Part A of Figure 4). Regarding the network structure, no significant difference was found between urban and rural adolescents (*M* = 0.13, *p* = 0.91, see Part B of Figure 4). When considering prospective and inhibitory anxiety in the network structure of depression, the global strength of rural adolescents (5.43) was significantly higher than that of urban adolescents (5.04), with a significant global strength difference (*S* = 0.38, *p* = 0.04), see Part C of Figure 4. Finally, there is no significant difference in network structure between urban and rural adolescents (*M* = 0.13, *p* = 0.91), see Part D of Figure 4.

### 3.5. Sensitivity Analysis

Although there were no significant differences in age and gender between rural and urban participants, given the disparity in group sizes, this study followed previous methodologies [44,45] and conducted covariate network analyses by including age and gender variables separately for each group. By comparing the network matrices without covariates to those with covariates, the results revealed significant similarities in network structures for both rural (*r* = 0.988, *p* < 0.001) and urban groups (*r* = 0.998, *p* < 0.001).

## 4. Discussion

### 4.1. Differences in the Prevalence Rate of and Mean Level of Depression between Urban and Rural Adolescents

In our study, the prevalence rate of depression among rural adolescents (40.1%) is significantly higher than that among their counterparts in urban areas (34.8%), which verified Hypothesis 1. Meanwhile, this finding coincides with an earlier study, which indicated that adolescents in rural areas face a higher risk of depression [12]. As aforementioned, the development level may differ greatly between urban and rural areas, which may lead to the inequality of socio-economic, policy, and cultural environment and mental support resources between these areas [8,11]. Meanwhile, parenting style may also divide urban and rural areas. A study suggested that rural parents had poorer parenting resources and reported higher levels of negative parenting (i.e., indicated by high levels of negative control, hostility, and rejection) [46]. Hence, poorer access to mental support services, as well as more negative parenting, may hinder the healthy development of rural adolescents and contribute to the occurrence of depression [47,48].

However, inconsistent with the current study, another study found that the prevalence rate of depression is slightly higher in urban adolescents (35.11%) than in rural adolescents (32.47%) and attributed the discrepancy to the greater academic burden that urban adolescents had to take [11]. It is worth noting that this study was conducted in Wuhan, Hubei Province, which is situated in central China and prioritizes economic development, even in rural regions. Consequently, the representativeness of the sample for adolescents with low socio-economic status (SES) in rural areas remains open to debate. Despite this limitation, it is necessary to conduct further investigations into factors such as academic burden, economic development, parenting styles, and their potential impacts on adolescents’ mental health. These factors may vary between urban and rural areas and warrant deeper exploration.

Meanwhile, the result of the *t*-test also suggested that rural adolescents obtained more severe symptoms of depression, including “sleep”, “guilt”, “concentration”, “motor”, and “suicide”, compared to their urban counterparts. This result provided further evidence for the implication that depression is more severe and deserves more attention in rural areas.

### 4.2. Network Comparison of Depression Symptom Network between Urban and Rural Adolescents

In the current study, we compared the symptom network of depression among adolescents from urban and rural areas. Inconsistent with our Hypothesis 2, the result of NCT indicated no significant difference between the global strength and network structure of these two networks. Moreover, the strongest edges and central symptoms also showed few discrepancies among the network of all adolescents, rural adolescents, and urban adolescents. The similarity of the network may be attributed to the fact that adolescents, regardless of various regions, shared similar developmental tasks that shaped adolescents’ unique mental characteristics [49]. Similarly, a recent study compared the depression network structure among junior students, senior students, college students, and the elderly in the same region and found the depression symptom network is much more alike in senior students, junior students, and college students than that in the elderly [50], which may further prove that development stage may greatly contribute to the symptom network structure.

However, the EI values of symptoms show some discrepancy, partly verifying Hypothesis 2. Specifically speaking, “motor” served as the symptom with the highest EI value in both rural and urban adolescents, and the EI value of “motor” among urban adolescents is higher than that among rural adolescents, which indicates that “motor” shared stronger connection with other depression symptoms, especially among urban adolescents. According to previous studies, in addition to cognitive and emotional disturbances of depression, “motor” is related to the neurobiological mechanism and may precisely predict increasing levels of depression [51]. Therefore, our findings suggested that the neurobiological changes due to depression deserve more attention, especially among urban adolescents.

### 4.3. Network Comparison of Flow Networks between Urban and Rural Adolescents

Moreover, we also compared the flow networks of rural and urban adolescents and found that the global strength of the flow network among rural adolescents is greater than that among urban adolescents, which verified Hypothesis 3a. The result may suggest that the relationship between IU and depression is significantly stronger among rural adolescents than their urban counterparts. Notably, adolescents in rural areas may grow up in a family with a lower SES than their urban peers [11]. Under lower SES, rural adolescents may have fewer available resources when confronting uncertainty, decreasing their ability to cope with ambiguous environments [52] and leading to increased negative feelings [53]. Apart from that, it is noticeable that some adolescents live in rural areas with one or both parents serving as migrant workers in urban areas; these adolescents are known as the left-behind children (LBC) [54]. Considering the lack of emotional support and normal family relationships, LBC are especially vulnerable to the impacts of uncertainty [55], which may also explain the closer relationship between IU and depression in rural adolescents.

Our study has identified some distinctions between the flow networks of adolescents from urban and rural areas. First, we observe that prospective anxiety and inhibitory anxiety are indirectly related to “appetite” among urban adolescents. In contrast, among rural adolescents, prospective anxiety is directly related to all depression symptoms, and inhibitory anxiety is only indirectly related to “anhedonia”. Consistent with our previous finding, this diversity may indicate that in rural adolescents, IU shares a stronger connection with depression.

Moreover, it is noticeable that “guilt” is strongly connected with both prospective anxiety and inhibitory anxiety in rural adolescents, but not in their urban peers. The unique connection between guilt and IU in rural adolescents could be attributed to the distinctive parenting styles [46]. An old saying in China states, ‘The children of the poor are heads of households’, suggesting that adolescents in poorer families are more expected to be independent early. Compared to parents in urban areas, parents in rural areas tend to provide less parental investments (i.e., parental expenditures on material and time that benefit their offspring) to their kids [56]. Under these conditions, adolescents may be more likely to experience guilt when they are expected to deal with ambiguous situations without sufficient support but fail to cope. Apart from that, parents in rural areas tend to use more corporal punishment in parenting practice [57], which may lead to internalizing problems in adolescents [58], which is strongly related to guilt [59].

Our study also found that in the flow networks among rural adolescents, “suicide ideation” is strongly connected to inhibitory anxiety but not to prospective anxiety. This finding aligns with our Hypothesis 3b that two components of IU may connect with depression symptoms differently. In the escape theory of suicide, suicide is viewed as the attempt to escape from painful thoughts and feelings about themselves and the situation [60]. Similarly, inhibitory anxiety is the avoidance tendency to escape from ambiguous situations [18]. As mentioned, rural adolescents have less access to family support [56]. Therefore, it is understandable that, when facing an uncertain situation without necessary family support, rural adolescents may view committing suicide as a way to escape from the fearful situation, which supports the former finding that the suicide rate is much higher in rural areas [61]. Hence, it should raise more attention that interventions aimed at improving rural adolescents’ ability to cope with uncertainty and providing more access to parental education for rural parents are in great need.

The study’s findings have important implications for addressing the increasing prevalence of depression among adolescents, especially in rural areas of China. The higher prevalence of depression in rural adolescents, along with the association between depression and IU, emphasizes the need for targeted interventions that focus on improving their coping skills and managing anxiety related to uncertainty. Identifying the direct association between “guilt” and prospective and inhibitory anxiety in rural adolescents’ flow networks provides valuable insights for designing effective early intervention strategies. Implementing culturally sensitive and community-based interventions is crucial to supporting the mental well-being of rural adolescents and fostering healthier communities.

## 5. Limitations

Several limitations should be mentioned in the current study. First, IU is closely related to anxiety and may contribute to the development of general anxiety disorder [27]. However, the current study did not explore the relationship between depression, general anxiety disorder (GAD), and IU. Therefore, further studies are needed to explore the interactions among IU, depression, and GAD in rural adolescents. Furthermore, the participants in our study ranged in age from 16 to 24 years, which covers middle and late adolescence. During adolescence, those in early adolescence are particularly vulnerable to stress [62], which may be different from those in middle and late adolescence. Therefore, studies that divided participants into more specific age groups to explore the possibly different relationship between depression and IU are warranted.

## 6. Conclusions

The current study found that the prevalence rate of depression in rural adolescents (40.1%) is significantly higher than that in their urban counterparts (34.8%). However, the depression networks show little discrepancy between these two groups. Additionally, we compared the flow networks of IU and depression in urban and rural adolescents and identified some differences. Firstly, the global strength of the flow network in rural adolescents is higher than in urban adolescents. Furthermore, in rural adolescents, “guilt” is strongly associated with both components of IU, whereas this relationship is not observed in urban adolescents. Moreover, in the flow network of rural adolescents, suicide ideation is strongly linked to inhibitory anxiety. On the other hand, in both urban and rural adolescents, “appetite” and “anhedonia” indirectly relate to inhibitory anxiety. These findings highlight the need for interventions aimed at improving coping mechanisms for uncertainty among rural adolescents to more effectively prevent and address their depressive symptoms.

## Figures and Tables

**Figure 1 behavsci-13-00662-f001:**
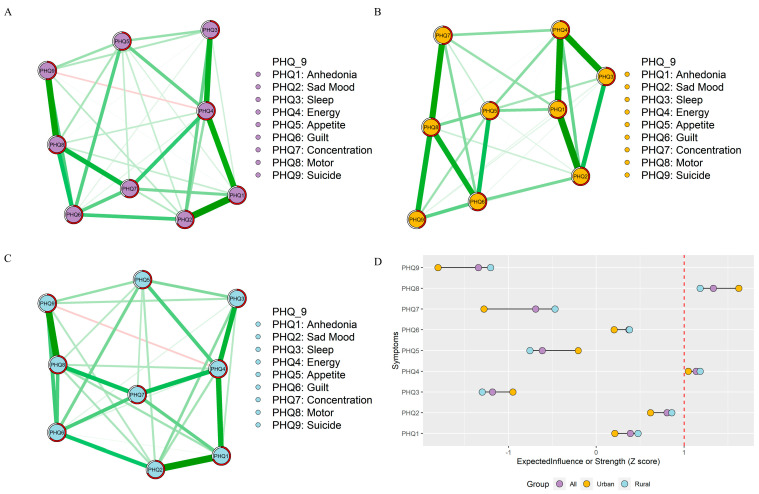
Network structures and centrality index. (**A**) Network structure of all adolescents. (**B**) Network of the urban adolescents. (**C**) Network of the rural adolescents. (**D**) The centrality value among all, urban, and rural adolescents (the red dotted line represents the EI of the strength of 1 (standardized)).

**Figure 2 behavsci-13-00662-f002:**
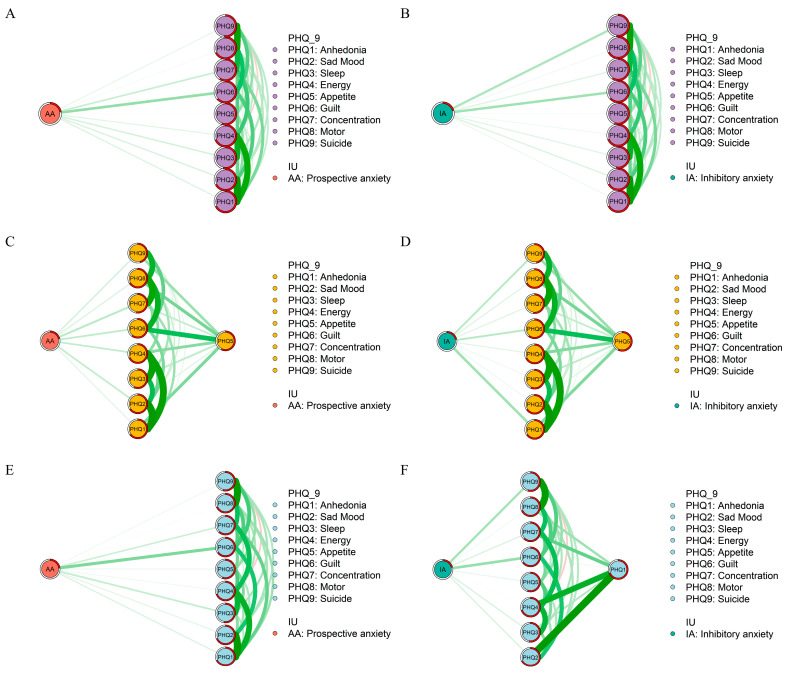
Flow Network structures. (**A**) Flow network structure between prospective anxiety and depression among all adolescents. (**B**) Flow network structure between inhibitory anxiety and depression among all adolescents. (**C**) Flow network structure between prospective anxiety and depression among urban adolescents. (**D**) Flow network structure between inhibitory anxiety and depression among urban adolescents. (**E**) Flow network structure between prospective anxiety and depression among rural adolescents. (**F**) Flow network structure between inhibitory anxiety and depression among rural adolescents.

**Figure 3 behavsci-13-00662-f003:**
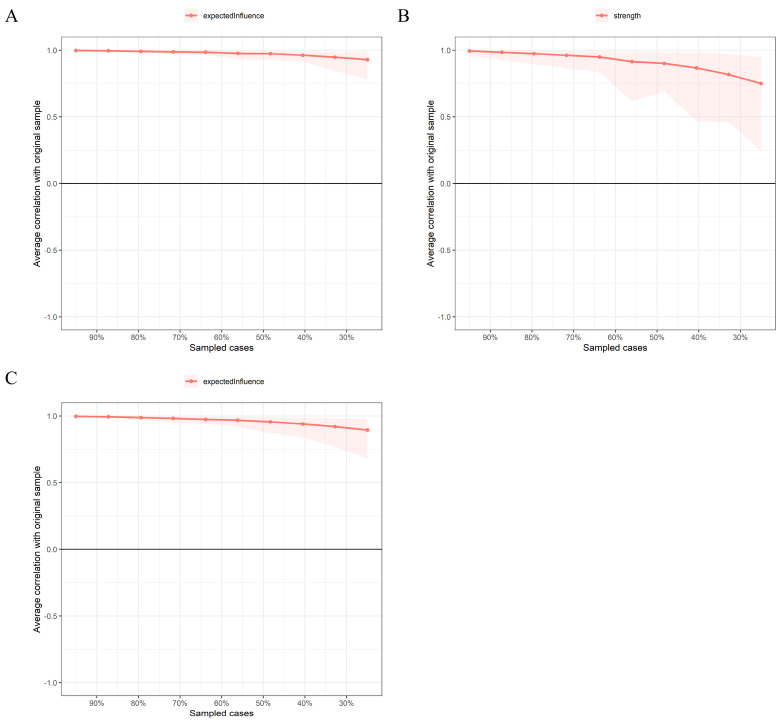
The x-axis indicates the percentage of cases of the original sample included at each step. The y-axis indicates the average of the correlations between the centrality indices from the original network and the centrality indices from the networks that were re-estimated after excluding increasing percentages of cases. (**A**) indicates all adolescents. (**B**) indicates urban adolescents. (**C**) indicates rural adolescents.

**Figure 4 behavsci-13-00662-f004:**
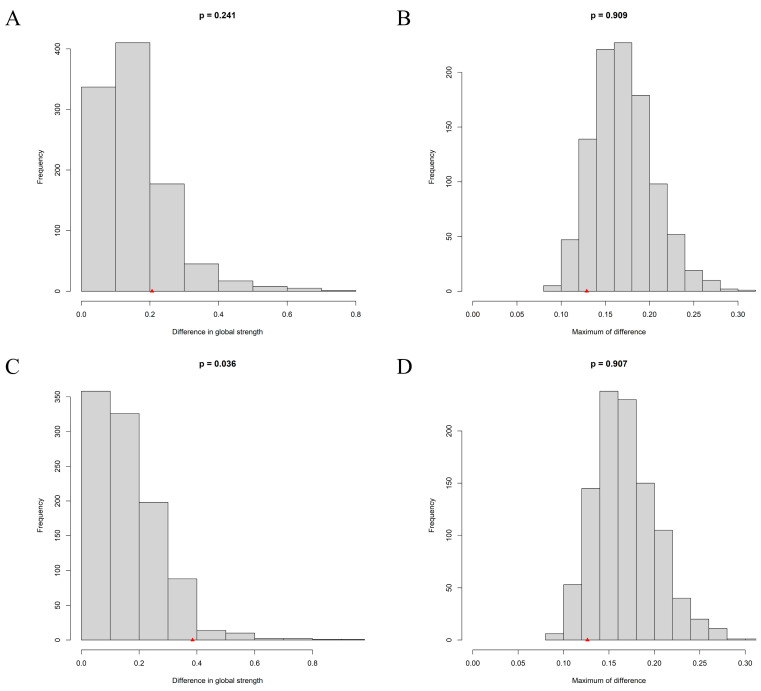
Network Comparison Test. (**A**) the difference in global strength for the depression network. (**B**) the maximum difference in edge strength for the depression network. (**C**) the difference in global strength for the flow network. (**D**) the maximum difference in edge strength for the flow network. The red triangle indicates the test statistic based on the observed (real) data.

**Table 1 behavsci-13-00662-t001:** Descriptive statistics for variables.

		City (N = 439)				Rural (N = 1049)				
Items	*Node*	*M*	*SD*	*Skewness*	*Kurtosis*	*M*	*SD*	*Skewness*	*Kurtosis*	*t*
PHQ1	Anhedonia	0.77	0.83	0.89	0.19	0.84	0.83	0.78	0.04	−1.41
PHQ2	Sad Mood	0.70	0.79	1.00	0.47	0.73	0.80	0.90	0.18	−0.65
PHQ3	Sleep	0.64	0.87	1.23	0.59	0.75	0.88	0.95	−0.03	−2.20 *
PHQ4	Energy	0.81	0.87	0.86	−0.04	0.88	0.87	0.75	−0.18	−1.46
PHQ5	Appetite	0.62	0.83	1.20	0.60	0.69	0.85	1.07	0.32	−1.47
PHQ6	Guilt	0.57	0.82	1.34	0.99	0.68	0.84	1.04	0.22	−2.39 *
PHQ7	Concentration	0.71	0.87	1.00	0.03	0.82	0.90	0.87	−0.14	−2.30 *
PHQ8	Motor	0.49	0.76	1.48	1.48	0.61	0.82	1.22	0.69	−2.61 **
PHQ9	Suicide	0.27	0.61	2.51	6.24	0.40	0.73	1.81	2.46	−3.30 **
AA	Prospective anxiety	2.54	0.79	0.36	0.10	2.54	0.83	0.56	0.28	0.05
IA	Inhibitory anxiety	2.41	0.82	0.50	0.07	2.40	0.86	0.70	0.35	0.03

Note. * indicates *p* < 0.05, ** indicates *p* < 0.01.

## Data Availability

Data are available upon request from the first author.

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
