# Peer review of "Comparing Depression Prevalence and Associated Symptoms with Intolerance of Uncertainty among Chinese Urban and Rural Adolescents: A Network Analysis"

_behavsci, 2023, doi:10.3390/bs13080662_

Round 1
Reviewer 1 Report
The article is of great value. I have suggestion only for minor revision.Living in rural areas could have advantages as well (e.g. nature relatedness), which should be given more focus in introduction.
Please provide 300 dpi jpg-s for figures.
Author Response
Dear reviewers,
Thank you for providing us with the opportunity to revise our manuscript titled: “Comparing the depression prevalence and symptoms association with intolerance of uncertainty among Chinese urban and rural adolescents: A network analysis”. We also greatly appreciate the reviewer’s suggestions and comments, which contribute a lot to the improvement of this paper. We are submitting the revised manuscript after changes are made in response to all the reviewers’ critiques.
Following is our point-to-point response to the reviewers’ comments. The modified contents are marked in red in the revised manuscript.
Please let us know if anything else is needed. We are willing to make further improvements if necessary. Thank you again for all the help.
Best wishes.
Comments and Suggestions for Authors
The article is of great value. I have suggestion only for minor revision. Living in rural areas could have advantages as well (e.g., nature relatedness), which should be given more focus in introduction.
Re: Thank you for your revision. It’s undoubtful that living in rural areas could have some advantages. However, the previous studies compared the mental health state and living condition between rural and urban adolescents mainly focused on the lower social-economic status of rural adolescents [1], the lack of sufficient parent support [2], and the less qualified education [3], which may hamper the health development of adolescents. Yet, the opinion that living in rural areas may bring advantages is very insightful and interesting, which deserve further exploration but is less relevant with the current study.
References
- Li, G.; Mei, J.; You, J.; Miao, J.; Song, X.; Sun, W.; Lan, Y.; Qiu, X.; Zhu, Z., Sociodemographic characteristics associated with adolescent depression in urban and rural areas of Hubei province: a cross-sectional analysis. BMC Psychiatry 2019, 19 (1), 386.
- Han, J.; Hao, Y. J.; Cui, N. X.; Wang, Z. H.; Lyu, P.; Yue, L., Parenting and parenting resources among Chinese parents with children under three years of age: rural and urban differences. Bmc Primary Care 2023, 24 (1).
- Wang, J. X.; Tigelaar, D. E. H.; Admiraal, W., Connecting rural schools to quality education: Rural teachers' use of digital educational resources. Computers in Human Behavior 2019, 101, 68-76.
Please provide 300 dpi jpg-s for figures.
Re: Thank you for your suggestions, previous figures have been replaced with figures of higher quality in the manuscript.

Reviewer 2 Report
Esteemed colleagues,
the topic you covered in the present study is valuable and highly important, from both clinical and educational points of view.
Still, I have the following comments regarding the manuscript:
- Depression - as a key concept of the manuscript, should be more detailed in the introduction and also in relation to adolescence;
- Also, the stress theory can be better documented, in respect to newer studies; (line 64)
- references should be added to the lines 95, 96 ... previous studies...
- network analysis is insufficiently conceptualized
For the Methods, the participants should be better detailed, as it follows:
- the participants are described as undergraduate students, enrolled in two applied universities, across all four years of studies, but the age range is indicated between 10 and 24 years. This issue should be clarified!
- the two groups which are analyzed - rural and urban - are disproportioned, how was that controlled in the analysis??
- how many females/males are in the two groups?
- given the developmental differences between the range 10-24, subgroups of ages should be generated and analyzed (Manova - age continuous variable, or other - cluster...)
- lines 159 and 167 Cronbach coefficient should be indicated in accordance with APA style
- lines 214 and 217 the brackets should be corrected
Concerning the discussion, how was the SES explored in the analysis, or the parenting style for being mentioned as possible explanation of the results
The limitations refer to adolescents between 10 and 24 years old.
Author Response
Dear reviewers,
Thank you for providing us with the opportunity to revise our manuscript titled: “Comparing the depression prevalence and symptoms association with intolerance of uncertainty among Chinese urban and rural adolescents: A network analysis”. We also greatly appreciate the reviewer’s suggestions and comments, which contribute a lot to the improvement of this paper. We are submitting the revised manuscript after changes are made in response to all the reviewers’ critiques.
Following is our point-to-point response to the reviewers’ comments. The modified contents are marked in red in the revised manuscript.
Please let us know if anything else is needed. We are willing to make further improvements if necessary. Thank you again for all the help.
Best wishes.
the topic you covered in the present study is valuable and highly important, from both clinical and educational points of view.
Still, I have the following comments regarding the manuscript:
- Depression - as a key concept of the manuscript, should be more detailed in the introduction and also in relation to adolescence;
Re: Thank you for your suggestion. We have added more discussion on how depression may influence the health development of adolescents as below:
Moreover, depression can hamper the healthy development of adolescents. According to a previous study, depression may lead to the impairment of the central executive process, leading to poorer academic performance [3]. Meanwhile, adolescents suffering from depression tend to show more problem behaviors and more social problems [4], and more alarmingly, depression was found to serve as an important risk factor for suicide attempts and suicide ideas among adolescents [5]. Indeed, to provide evidence for designing effective interventions, it is crucial to further explore the underlying mechanism of depression among adolescents.
References
- Owens, M.; Stevenson, J.; Hadwin, J. A.; Norgate, R., Anxiety and depression in academic performance: An exploration of the mediating factors of worry and working memory. School Psychology International 2012, 33 (4), 433-449.
- Cermak, I.; Klimusova, H.; Vizdalova, H., Depression in childhood and its relation to problems of behaviour. Ceskoslovenska Psychologie 2005, 49 (3), 223-236.
- Wolfersdorf, M., Depression and suicide. Bundesgesundheitsblatt-Gesundheitsforschung-Gesundheitsschutz 2008, 51 (4), 443-450.
- Also, the stress theory can be better documented, in respect to newer studies; (line 64)
Re: Thank for your suggestion, the stress theory has been better documented and newer studies has also been added as below:
According to the stress theory proposed by Lazarus [19], stress is defined as the interaction between individuals and their environment, and stress may appear when a person perceives the environment as harmful. This suggests that the experience of stress is heavily influenced by how people interpret the situations they encounter. Therefore, since people with higher levels of IU may tend to overestimate the likelihood of negative events when confronting uncertainty, they are more likely to become more stressed when confronting unpredictable situations [15, 20].
References
- Shihata, S.; McEvoy, P. M.; Mullan, B. A.; Carleton, R. N. J. J. o. a. d., Intolerance of uncertainty in emotional disorders: What uncertainties remain? 2016, 41, 115-124.
- Lazarus. (1991). Emotion and adaptation. Oxford University Press.
- Bakioglu, F.; Korkmaz, O.; Ercan, H., Fear of COVID-19 and Positivity: Mediating Role of Intolerance of Uncertainty, Depression, Anxiety, and Stress. International Journal of Mental Health and Addiction 2021, 19 (6), 2369-2382.
- references should be added to the lines 95, 96 ... previous studies...
Re: thank you for your suggestion. The references have been added to the lines as below.
Although previous studies have proved that IU is closely related to depression [15, 16, 20, 22], they failed to identify how two components of IU interact with different depression symptoms specifically.
References
- Shihata, S.; McEvoy, P. M.; Mullan, B. A.; Carleton, R. N. J. J. o. a. d., Intolerance of uncertainty in emotional disorders: What uncertainties remain? 2016, 41, 115-124.
- McEvoy, P. M.; Mahoney, A. E. J., To Be Sure, To Be Sure: Intolerance of Uncertainty Mediates Symptoms of Various Anxiety Disorders and Depression. Behavior Therapy 2012, 43 (3), 533-545.
- Bakioglu, F.; Korkmaz, O.; Ercan, H., Fear of COVID-19 and Positivity: Mediating Role of Intolerance of Uncertainty, Depression, Anxiety, and Stress. International Journal of Mental Health and Addiction 2021, 19 (6), 2369-2382.
- Korkmaz, H.; Guloglu, B., The role of uncertainty tolerance and meaning in life on depression and anxiety throughout Covid-19 pandemic. Personality and Individual Differences 2021, 179.
- network analysis is insufficiently conceptualized
Re: Thank you for your suggestion. We have added more description to better conceptualize the network analysis as below:
Compared to the traditional perspective, network analysis assumes that mental disor-ders can be defined as the combination of various symptoms and symptoms may be triggered by other symptoms [26]. Therefore, by analyzing the correlation between dif-ferent symptoms, the network analysis could identify the most influential symptom, which offers evidence for designing more effective interventions [26].
References
- Borsboom., A network theory of mental disorders. World Psychiatry 2017, 16(1), 5–13.
For the Methods, the participants should be better detailed, as it follows:
- the participants are described as undergraduate students, enrolled in two applied universities, across all four years of studies, but the age range is indicated between 10 and 24 years. This issue should be clarified!
Re: Thank you for your suggestion. We apologize for the misunderstanding caused by our description and we have better described the participants as below:
In the present study, a written questionnaire was administered to undergraduate students enrolled in two applied universities in the western region of China in April 2023. The sample comprised students across all four years of study. A total of 1520 questionnaires were gathered. After considering that certain participants’ data on age did not align with the typical youth age range (i.e., 10–24 years old, [29]), a total of 32 data points were eliminated (i.e., There are 15 participants without age information, 15 participants over 24 years old, and two participants whose age is given as 2 years old.). The final data analysis comprised a total of 1488 questionnaires (female = 640; Mean age = 20, SD age = 1.51, range age 16-24). According to the students’ household registration, the students were divided into urban (N = 439, female, 198; Mean age = 20, SD age = 1.54, range age 17-24) and rural (N = 1049, female, 442; Mean age = 20, SD age = 1.50, range age 16-24) groups. Considering the difference in population size between urban and rural areas, t-tests and Chi-square tests were conducted to examine the differences in age and gender between participants in the two groups. The results indicated that there were no significant differences in age (t = 0.49, p = 0.623, Cohen’s d = 0.03) and gender (c2 = 1.11, p = 0.29) between the two groups.
- the two groups which are analyzed - rural and urban - are disproportioned, how was that controlled in the analysis?
Re: Thank you for your question. Undoubtfully, the disproportioned sample sizes of urban and rural adolescents may influence the reliability of the current study. Indeed, we utilized t-test and Chi-square test to examine that there exist no significant differences in the gender and ages of the two groups. Moreover, we also conducted covariate network analyses by including age and gender variables separately for each group and found that no significant difference exist in the network structure for both groups. Detailed are as below:
Considering the difference in population size between urban and rural areas, t-tests and Chi-square tests were conducted to examine the differences in age and gender between participants in the two groups. The results indicated that there were no significant differences in age (t = 0.49, p = 0.623, Cohen’s d = 0.03) and gender (c2 = 1.11, p = 0.29) between the two groups.
3.5 Sensitivity analysis
Although there were no significant differences in age and gender between rural and urban participants, given the disparity in group sizes, this study followed previous methodologies [44, 45] and conducted covariate network analyses by including age and gender variables separately for each group. By comparing the network matrices without covariates to those with covariates, the results revealed significant similarities in network structures for both rural (r = 0.988, p < .001) and urban groups (r = 0.998, p < .001).
References
- Tao, Y., Niu, H., Hou, W., Zhang, L., & Ying, R., Hopelessness during and after the COVID-19 pandemic lockdown among Chinese college students: A longitudinal network analysis. Journal of clinical psychology (2023), 79 (3), 748–761.
- Tao, Y., Wang, S., Tang, Q., Ma, Z., Zhang, L., & Liu, X., Centrality depression–anxiety symptoms linked to suicidal ideation among depressed college students––A network approach. PsyCh Journal 2023, 1–11.
- how many females/males are in the two groups?
Re: Thank you for your suggestion. We have added the information about the number of male and female in two groups as below:
According to the students’ household registration, the students were divided into urban (N = 439, female, 198; Mean age = 20, SD age = 1.54, range age 17-24) and rural (N = 1049, female, 442; Mean age = 20, SD age = 1.50, range age 16-24) groups.
- given the developmental differences between the range 10-24, subgroups of ages should be generated and analyzed (Manova - age continuous variable, or other - cluster...)
Re: Thank you for your suggestions. We apologize for the misunderstanding caused by our description. In fact, the ages the participants in our study range from 16-24, which only covered the late adolescence. Therefore, we believe that there’s less necessity to divide subgroups of ages.
- lines 159 and 167 Cronbach coefficient should be indicated in accordance with APA style
Re: Thank you for your suggestions. We have changed the Cronbach coefficient in accordance with APA style as below:
In the present study, PHQ-9 was found to be highly reliable (α = .94).
The Chinese version of the IUS-12 has demonstrated good reliability and validity [34], with high reliability (α = .89) in the present study
- lines 214 and 217 the brackets should be corrected
Re: Thank you for your suggestions, the brackets have been corrected as below:
We utilized the NetworkComparisonTest package (version 2.2.1) in R, developed by van Borkulo et al. [42], to perform the network comparison test (NCT).
References
- Van Borkulo, C. D.; Van Bork, R.; Boschloo, L.; Kossakowski, J. J.; Tio, P.; Schoevers, R. A.; Borsboom, D.; Waldorp, L. J., Comparing Network Structures on Three Aspects: A Permutation Test. Psychological Methods 2022.
Concerning the discussion, how was the SES explored in the analysis, or the parenting style for being mentioned as possible explanation of the results
Re:Thank you for your suggestion. We divided the participants into two groups according to their household registration and individuals who registered in rural areas were viewed to have lower SES in average since in China, rural areas is less developed compared to the urban areas [1]. Meanwhile, previous study suggested that the parenting style and parental resources varied in rural and urban areas [2], which can influence the development of adolescents [3]. Indeed, we use parenting style as one possible explanation for the results.
- Zhang, L. In Research on Financial Support for Rural Economic Development in China, 3rd International Conference on Advanced Engineering Materials and Architecture Science (ICAEMAS), Huhhot, PEOPLES R CHINA, Jul 26-27; Huhhot, PEOPLES R CHINA, 2014; pp 1623-1626.
- Han, J.; Hao, Y. J.; Cui, N. X.; Wang, Z. H.; Lyu, P.; Yue, L., Parenting and parenting resources among Chinese parents with children under three years of age: rural and urban differences. Bmc Primary Care 2023, 24 (1).
- Florean, I. S.; Dobrean, A.; Roman, G. D., Early Adolescents’ Perceptions of Parenting Practices and Mental Health Problems: A Network Approach. Journal of Family Psychology 2022, 36 (3), 438-448.
The limitations refer to adolescents between 10 and 24 years old.
Re: Thank you for your suggestions. We are sorry about the mistake of age range in the limitations. We have corrected the limitation as below:
Furthermore, the participants in our study ranged in age from 16 to 24 years, which covers middle and late adolescence. During adolescence, early adolescents are particularly vulnerable to stress [62], which may be different from middle and late adolescents.
References
- Holder, M. K.; Blaustein, J. D., Puberty and adolescence as a time of vulnerability to stressors that alter neurobehavioral processes. Frontiers in Neuroendocrinology 2014, 35 (1), 89-110.

Reviewer 3 Report
- Format of references is not based on journal.
- The quality of pictures is very poor and not understandable.
- Please add some practical implications for the findings of this paper.
Author Response
Dear reviewers,
Thank you for providing us with the opportunity to revise our manuscript titled: “Comparing the depression prevalence and symptoms association with intolerance of uncertainty among Chinese urban and rural adolescents: A network analysis”. We also greatly appreciate the reviewer’s suggestions and comments, which contribute a lot to the improvement of this paper. We are submitting the revised manuscript after changes are made in response to all the reviewers’ critiques.
Following is our point-to-point response to the reviewers’ comments. The modified contents are marked in red in the revised manuscript.
Please let us know if anything else is needed. We are willing to make further improvements if necessary. Thank you again for all the help.
Best wishes.
Comments and Suggestions for Authors
- Format of references is not based on journal.
Re: Thank you for your suggestion. We apologize for the incorrected format and have changed the format according to the requestion of the journal.
- The quality of pictures is very poor and not understandable.
Re: Thank you for your suggestion. We have replaced the previous pictures with higher-quality pictures.
- Please add some practical implications for the findings of this paper.
Re: Thank you for your suggestions. We have added some practical implications for the findings of the current study in the end of the discussion as below:
The study's findings have important implications for addressing the increasing prevalence of depression among adolescents, especially in rural areas of China. The higher prevalence of depression in rural adolescents, along with the association be-tween depression and IU, emphasizes the need for targeted interventions that focus on improving their coping skills and managing anxiety related to uncertainty. Identifying the direct association between "guilt" and prospective and inhibitory anxiety in rural adolescents' flow networks provides valuable insights for designing effective early in-tervention strategies. Implementing culturally sensitive and community-based inter-ventions is crucial to supporting the mental well-being of rural adolescents and fos-tering healthier communities.

Reviewer 4 Report
This study looked into the prevalence rate of depression among adolescents in rural and urban areas. In addition, they examined the various possible network structures of depression in urban and rural adolescents through symptom networks. Their last aim was to explore the interaction between the two factors if Intolerance of Uncertainty and depression among their sample using the flow network approach. It is interesting study, but I have few comments/concerns:
· Procedure: when was data collected? Inclusion/exclusion criteria
· Participants:
o Did you control for cases of participants whose current residence is different from the household registration? For example, one may be registered in the rural area, but live temporarily due to the university duties in the urban area.
What was the age range of your sample
o Did you consider large difference in sample size between your groups? How did you control for it? What was the gender ratio in each of the groups? Were there differences in the mean age of two groups?
· How did you decide to refer to your sample as adolescents considering that they were from 10 to 24 years old? Reference you used (McDonagh et al., 2018) defines adolescence from 10-19 and young adulthood 20-24.
· Bad resolution of all figures. It is completely unreadable.
Author Response
Dear reviewers,
Thank you for providing us with the opportunity to revise our manuscript titled: “Comparing the depression prevalence and symptoms association with intolerance of uncertainty among Chinese urban and rural adolescents: A network analysis”. We also greatly appreciate the reviewer’s suggestions and comments, which contribute a lot to the improvement of this paper. We are submitting the revised manuscript after changes are made in response to all the reviewers’ critiques.
Following is our point-to-point response to the reviewers’ comments. The modified contents are marked in red in the revised manuscript.
Please let us know if anything else is needed. We are willing to make further improvements if necessary. Thank you again for all the help.
Best wishes.
Comments and Suggestions for Authors
This study looked into the prevalence rate of depression among adolescents in rural and urban areas. In addition, they examined the various possible network structures of depression in urban and rural adolescents through symptom networks. Their last aim was to explore the interaction between the two factors if Intolerance of Uncertainty and depression among their sample using the flow network approach. It is interesting study, but I have few comments/concerns:
Procedure: when was data collected? Inclusion/exclusion criteria
Re: Thank you for your suggestions. The exclusion criteria has been added in the methods part as below:
A total of 1520 questionnaires were gathered. After considering that certain participants’ data on age did not align with the typical youth age range (i.e., 10–24 years old, [29]), a total of 32 data points were eliminated (i.e., There are 15 participants without age information, 15 participants over 24 years old, and two participants whose age is given as 2 years old.).
References
- Sawyer, S. M., Azzopardi, P. S., Wickremarathne, D., & Patton, G. C., The age of adolescence... and young adulthood. Lancet Child & Adolescent Health 2018, 2(4), e6-e6.
Participants:
Did you control for cases of participants whose current residence is different from the household registration? For example, one may be registered in the rural area, but live temporarily due to the university duties in the urban area.
Re: Thank you for your suggestions. In the process of data collecting, all participants were asked to report their household registration, and we divided the participants into rural and urban groups according to their reported household registration. Therefore, even if rural students moved to urban areas due to university duties, they still belong to the rural groups since their household registration is in rural areas.
What was the age range of your sample
Re: Thank you for your suggestions. The age range of our sample is 16 to 24, which has been clarified in the paper below:
The final data analysis comprised a total of 1488 questionnaires (female, 640; Mean age = 20, SD age = 1.51, range age=16-24).
Did you consider large difference in sample size between your groups? How did you control for it? What was the gender ratio in each of the groups? Were there differences in the mean age of two groups?
Re: Thank you for your suggestions. We utilized a t-test and Chi-square test to examine that there exist no significant differences in the gender and ages of the two groups. Moreover, we also conducted covariate network analyses by including age and gender variables separately for each group and found that no significant difference existed in the network structure for both groups. Details are as below:
Considering the difference in population size between urban and rural areas, t-tests and Chi-square tests were conducted to examine the differences in age and gender between participants in the two groups. The results indicated that there were no significant differences in age (t = 0.49, p = 0.623, Cohen’s d = 0.03) and gender (c2 = 1.11, p = 0.29) between the two groups.
3.5 Sensitivity analysis
Although there were no significant differences in age and gender between rural and urban participants, given the disparity in group sizes, this study followed previous methodologies [48] and conducted covariate network analyses by including age and gender variables separately for each group. By comparing the network ma-trices without covariates to those with covariates, the results revealed significant similarities in network structures for both rural (r = 0.988, p < .001) and urban groups (r = 0.998, p < .001).
- How did you decide to refer to your sample as adolescents considering that they were from 10 to 24 years old? Reference you used (McDonagh et al., 2018) defines adolescence from 10-19 and young adulthood 20-24.
Re: Thank you for your suggestions. We apologize for the misunderstanding caused by our description about the age range of our participants. In fact, the ages of our participants range form 16 to 24, covering the most part of youth, as stated by Sawyer [1].
[1] Sawyer, S. M., Azzopardi, P. S., Wickremarathne, D., & Patton, G. C., The age of adolescence... and young adulthood. Lancet Child & Adolescent Health 2018, 2(4), e6-e6.
- Bad resolution of all figures. It is completely unreadable.
Re: Thank you for your suggestions. Figures have replaced the previous figures with higher quality in the paper.

Round 2
Reviewer 4 Report
Thank you for answering all my questions and concerns. Paper has been improved now, and I endorse its publication. Congratulations on the good job.